# Impact of Dietary Patterns on Skeletal Health: A Systematic Review and Meta-Analysis of Bone Mineral Density, Fracture, Bone Turnover Markers, and Nutritional Status

**DOI:** 10.3390/nu17243845

**Published:** 2025-12-09

**Authors:** Adhithya Mullath Ullas, Joseph Boamah, Amir Hussain, Ioanna Myrtziou, Ioannis Kanakis

**Affiliations:** 1Chester Medical School, Faculty of Health, Medicine and Society, University of Chester, Chester CH1 4BJ, UK; adhithyamullath@gmail.com (A.M.U.); oxfordwint@yahoo.com (J.B.); dr.hussainamir@gmail.com (A.H.); 2Department of Musculoskeletal & Ageing Science, Institute of Life Course & Medical Sciences (IL-CaMS), University of Liverpool, Liverpool L7 8TX, UK

**Keywords:** Mediterranean diet, calorie restriction diet, high-protein diet, low-carbohydrate/ketogenic diet, bone mineral density, fracture, bone health

## Abstract

**Background/Objectives:** Dietary patterns play a crucial role in musculoskeletal health; however, the effects of different diets on bone mineral density (BMD), fracture risk, and bone metabolism remain inconsistent across studies. This systematic review and meta-analysis aimed to evaluate the impact of Mediterranean, calorie restriction, high-protein, low-carbohydrate, and ketogenic diets on skeletal outcomes in adults. **Methods:** A comprehensive search of PubMed/MEDLINE, CENTRAL, and Web of Science was conducted for studies published between January 2000 and June 2025. Eligible randomised controlled trials (RCTs) and cohort studies involving adults (≥18 years) and reporting outcomes related to BMD, fractures, bone turnover markers, and vitamin D or calcium status were included. Risk of bias was assessed using the Cochrane’s Risk of Bias tool for RCTs and the Joanna Briggs Institute checklist for observational studies. Random-effects meta-analyses were performed for outcomes reported by ≥3 comparable studies, presenting standardised mean differences (SMDs) for BMD and hazard ratios (HRs) for fractures. **Results:** Thirty studies met inclusion criteria, comprising 14 RCTs and 16 observational studies with over 500,000 participants. Pooled analyses showed no significant differences in BMD at the femoral neck (SMD = 0.12, 95% CI −0.80 to 1.04), lumbar spine (SMD = 0.04, 95% CI: −1.12 to 1.03), total hip (SMD = −0.07, 95% CI −0.36 to 0.21), or whole body (SMD = 0.03, 95% CI −0.07 to 0.14) across diet categories. However, adherence to a Mediterranean diet was associated with a significantly reduced hazard of hip and overall fractures (pooled HR = 0.95, 95% CI 0.93–0.96). Calorie restriction consistently increased bone resorption markers, whereas Mediterranean and high-protein diets showed neutral or modestly favourable effects. Vitamin D and calcium status were minimally affected across interventions. **Conclusions:** While dietary patterns exert diverse effects on skeletal health, consistent evidence supports Mediterranean-style diets as protective against fractures. Calorie restriction may elevate bone turnover, whereas ketogenic and high-protein diets show mixed effects on bone. However, across all analyses, high heterogeneity was observed. Further high-quality RCTs are warranted to clarify these relationships and inform dietary guidance for bone health.

## 1. Introduction

Musculoskeletal (MSK) diseases represent a significant and growing public health concern worldwide, contributing to decreased mobility, disability, and reduced quality of life among ageing populations [1]. Conditions such as osteoporosis, osteopenia, fragility fractures, sarcopenia, and falls are closely interrelated, often coexisting and compounding risk for adverse health outcomes [2]. Osteoporosis and osteopenia are characterised by weak bones and low bone mineral density (BMD) with increased fracture risk [3,4]; sarcopenia involves progressive loss of skeletal muscle mass and strength [5]. Fragility fractures, particularly of the hip, vertebrae, and wrist, are among the most devastating consequences, leading to chronic pain and loss of independence [6]. Collectively, these conditions pose a significant global health burden, with osteoporosis affecting an estimated 200 million people worldwide [7]. Among individuals over the age of 50, approximately one in three women and one in six men will experience an osteoporotic fracture during their lifetime [8]. Hip fractures alone account for millions of disability-adjusted life years (DALYs) annually, ranking among the leading causes of morbidity in older adults [9]. Economically, healthcare costs are substantial. Based on data from 2019, osteoporosis and fragility fractures impose an annual financial burden exceeding €56 billion on European healthcare systems [8].

The conventional management of musculoskeletal disorders is largely centred on pharmacological interventions [10]. Bisphosphonates, selective oestrogen receptor modulators, parathyroid hormone analogues, and denosumab are widely prescribed to improve BMD and reduce fracture risk [11,12], while vitamin D and calcium supplementation remain the cornerstone of preventive strategies [13]. Despite advances, these therapeutic approaches are not without significant limitations. Pharmacological treatments are often costly, posing barriers to widespread accessibility, particularly in low- and middle-income countries [14]. Long-term adherence remains a persistent clinical challenge, with evidence highlighting high rates of treatment discontinuation due to complexity of medication regimens, and suboptimal patient engagement [15]. Moreover, adverse effects such as gastrointestinal intolerance with bisphosphonates, hypocalcaemia with denosumab, or cardiovascular risks associated with certain agents further limit their long-term acceptability [16,17]. These challenges undermine both treatment initiation and long-term adherence, diminishing population-level efficacy of pharmacological interventions. While pharmacological treatments remain central to the management of musculoskeletal pain, growing attention has turned to complementary and non-pharmacological strategies, such as lifestyle interventions, particularly diet, as modifiable factors influencing bone, muscle, and joint health [18]. These strategies not only address the multifactorial determinants of MSK health but also provide sustainable, cost-effective options to manage these conditions [18,19].

Nutrition plays a pivotal role in skeletal health by acting through multiple pathways that extend beyond the provision of isolated nutrients. While calcium, vitamin D, and protein are fundamental to bone health and maintenance, dietary patterns overall influence musculoskeletal outcomes by integrating a broad spectrum of bioactive compounds, micronutrients, and macronutrient interactions [20,21]. Various types of diets exert differential influences on bone health. The Mediterranean diet, rich in fruits, vegetables, whole grains, legumes, fish, olive oil, and moderate wine consumption; the ketogenic diet, characterised by markedly reduced carbohydrates, accompanied by a moderate to high consumption of dietary fats, with the aim of inducing a metabolic state ketosis; the caloric restriction diet, defined as a diet with reduced caloric intake; and high-protein diets all provide benefits that extend beyond bone density to encompass bone quality, muscle strength, and reduced fracture risk [22,23]. These diets offer practical guidance that is easier for individuals to adopt and adhere to, leading to more actionable recommendations for chronic disease prevention and management. Emerging research suggests that adherence to the Mediterranean diet is associated with reduced systemic inflammation, improved bone turnover, and better functional outcomes [24,25,26]. Evidence-based dietary patterns have emerged as pivotal in shaping MSK health, with research linking specific nutritional profiles to improved BMD, muscle integrity, and reduced systemic inflammation [27,28,29]. These benefits arise from synergistic effects of nutrients, antioxidants, and anti-inflammatory components present in whole foods, which cannot be replicated by single-nutrient supplementation. However, the findings remain inconsistent, partly due to heterogeneity in study design, populations, and outcome measures.

Previous studies focus on single diets in a specific population with no integrated synthesis across dietary patterns. Given the ageing global population and the rising prevalence of MSK disorders, a comprehensive evaluation of dietary patterns is necessary. As a foundational element of daily living, diet continues to be central focus in efforts to optimise MSK outcomes. Synthesising current evidence through systematic review and meta-analysis will help clarify the extent to which various dietary approaches affect MSK health. In this work, we aim to systematically review and meta-analyse the effects of Mediterranean, ketogenic, high-protein, and calorie restriction diets on bone health outcomes, specifically BMD and fracture. To our knowledge, this is the first systematic review which collectively analyses the impact of these specific diets on bone homeostasis with a high number of participants.

## 2. Materials and Methods

This study was conducted in accordance with the Preferred Reporting Items for Systematic Reviews and Meta-Analyses (PRISMA) statement guidelines [30].

### 2.1. Search Strategy

A systematic search of the literature was performed utilising PubMed/MEDLINE, the Cochrane Central Register of Controlled Trials (CENTRAL), and Web of Science for studies published in English from January 2000 to June 2025. The search strategy was structured using the Population, Intervention, Comparison, and Outcomes (PICO) framework (Table 1). Search terms were derived from Medical Subject Headings (MeSH) terms, keywords, and synonyms, combined with Boolean operators (AND/OR). No restrictions were applied to study setting or follow-up duration. Filters were applied for randomised controlled trials (RCTs) and cohort studies in humans. All references were exported to Rayyan software (https://help.rayyan.ai/hc/en-us/articles/4406419348369-What-is-the-version-of-Rayyan, accessed on 13 September 2025) for duplication and screening [31]. The review was not registered.

### 2.2. Study Selection

All retrieved articles were screened using a two-phase process: (1) title and abstract screening, and (2) full-text review. Studies were eligible for inclusion if they (i) included male and female adults (≥18 years); (ii) examined at least one dietary pattern of interest (Mediterranean, ketogenic, high-protein, or calorie restriction); (iii) reported outcomes on BMD, fracture incidence, falls, bone turnover markers, vitamin D or calcium status, or lean body mass; and (iv) were designed as RCTs or prospective cohort studies. The exclusion criteria were as follows: (i) studies on children and adolescents (<18 years); (ii) case reports, reviews, letters, editorials, and conference abstracts; (iii) non-human (animal or in vitro) studies; (iv) interventions where the diet was combined with another supplement or pharmacological treatment, making independent assessment of diet impossible; and (v) studies without full-text availability or incomplete/unreliable data.

### 2.3. Reviewing Process and Data Extraction

Three independent reviewers (A.M.U., J.B., and A.H.) screened titles and abstracts, followed by full-text review up until September 2024. Discrepancies were resolved by consensus or consultation with a fourth reviewer. A standardised data extraction sheet was developed to extract detailed information in terms of study characteristics (author, year, country, study design), participants characteristics (sample size, sex, mean age, baseline health status), intervention details (diet type, duration, comparator), outcomes measured, and main results. All extracted data were cross-checked for accuracy.

### 2.4. Risk of Bias Evaluation

Risk of bias for all included RCTs was assessed using the Cochrane Collaboration’s Risk of Bias (RoB) tool [32]. Each RCT was assessed across five main domains: sequence generation, allocation concealment, blinding of participants/personnel/outcomes assessors, incomplete outcome data, selective reporting, and other biases. Each study was classified as “low risk”, high risk”, or “unclear risk”. All other studies were assessed using the Joanna Briggs Institute (JBI) critical appraisal checklist [33]. Multiple domains are assessed by this checklist, including appropriateness of the sampling frame, adequacy of sample size, representativeness of the study population, validity and reliability of the measurement methods, clarity of case definitions, and adequacy of statistical analysis.

### 2.5. Data Synthesis and Statistical Analysis

Quantitative synthesis was undertaken for outcomes reported by at least three comparable studies. Separate meta-analyses were performed for each bone site (femoral neck, lumbar spine, total hip, and whole body) and for fracture outcomes, stratified by dietary pattern. Pooled effect sizes for continuous outcomes (bone mineral density, BMD) were expressed as standardised mean differences (SMDs) with corresponding 95% confidence intervals (CIs). For fracture outcomes, pooled hazard ratios (HRs) with 95% CIs were calculated.

Random-effects models were used throughout to account for expected clinical and methodological heterogeneity across studies differing in population characteristics, dietary definitions, and follow-up duration. Statistical heterogeneity was evaluated using the χ^2^ test and quantified with the *I*^2^ statistic and τ^2^ estimate. Subgroup analyses were conducted according to dietary pattern (Mediterranean, calorie restriction, low-carbohydrate/ketogenic, and high-protein) to explore sources of heterogeneity. When fewer than three studies contributed to a comparison or when data were not comparable across designs, findings were summarised narratively.

Meta-analyses were conducted using the Cochrane Review Manager (RevMan, version 5.4) software. For each analysis, between-group differences were considered statistically significant at *p <* 0.05. Tests for subgroup differences were used to examine variation between diet categories. Forest plots were generated to visualise pooled estimates for each skeletal site and fracture outcome. To evaluate potential publication bias, funnel plots were constructed to analyse asymmetry for outcomes included ≥10 studies (femoral neck BMD, lumbar spine BMD, and fracture risk), according to the Cochrane Handbook for conducting Systematic Reviews (https://training.cochrane.org/handbook/current/chapter-13) (accessed on 14 October 2025).

For bone turnover markers, vitamin D, and calcium outcomes, where data were heterogeneous in measurement and reporting units, results were synthesised descriptively. No formal meta-analysis was performed for these variables due to inconsistent reporting across studies.

## 3. Results

### 3.1. Search Results

The PRISMA flow diagram schematically presents the searching process for each database search and selection of publications for review and meta-analysis (Figure 1). A total of 877 records were identified through electronic database searches (PubMed = 299; Cochrane 578), where all articles from Web of Science were duplicates. After removing 138 duplicate records using Rayyan software, 739 unique records remained for screening. Based on title and abstract, 613 records were excluded for not meeting the inclusion criteria. The remaining 126 full texts were sought for retrieval, of which 14 were inaccessible. Finally, 112 articles were assessed in full for eligibility. After full-text review, 82 studies were excluded with the following reasons: no relevant skeletal outcomes (n = 46), studies not involving dietary exposures or interventions (n = 8), studies on paediatric populations (n = 18), and those with incomplete or non-extractable data (n = 10). Consequently, 31 studies were included in the systematic review and meta-analysis.

### 3.2. Characteristics of Included Studies

A total of 30 studies investigating the relationship between dietary patterns and skeletal outcomes were included. The evidence base comprised 14 RCTs [28,34,35,36,37,38,39,40,41,42,43,44,45,46], nine prospective cohort studies [27,47,48,49,50,51,52,53,54], five cross-sectional analyses [55,56,57,58,59], one case–control study [60], and one longitudinal cohort [61]. Sample sizes ranged widely from n = 21 in small, controlled feeding studies [39,44] to n = 188,795 in large population cohorts [48], with a cumulative sample exceeding half a million participants across all included reports. Participant ages varied from young, resistance-trained adults [55] to frail older adults [34], with the majority of the studies including participants with an age > 50, and sex distribution differed by study design, with several including only postmenopausal women [50,58] or older men [27]. The studies represented a broad geographical distribution, including North America [34,36,53], Europe [37,45,48], Asia [56,58,60], and Australia [35,41]. Most studies recruited healthy adults or older adults from the general population, without major chronic disease at baseline, while a few studies included participants with overweight or obesity [34,35,38,39,41,42]. Vázquez-Lorente et al. [45] included participants with metabolic syndrome.

The interventions evaluated a variety of dietary patterns linked to bone health. Mediterranean-style diets were the most commonly assessed dietary pattern [37,45,47,49,50], followed by low-carbohydrate or ketogenic diets [35,39,41,44] and energy-restricted or high-protein weight-loss regimens [34,42,43,46]. Other interventions included nut- and olive oil-enriched diets [37,40], walnut supplementation [40], and adherence to healthy-eating indices or dietary-quality scores [57,60]. The durations for the interventions ranged from 8 weeks in short dietary trials [41,44] to nearly 16 years of follow-up in large observational cohorts [53]. The type of comparator also varied: several trials compared different macronutrient compositions (e.g., low-carbohydrate vs. low-fat [39]) or lifestyle components (e.g., MedDiet + exercise vs. usual diet [45]), while others used population quintiles of dietary-pattern adherence [47]. Primary outcomes across studies included bone mineral density (BMD) at lumbar spine, femoral neck, hip, or whole body [28,35,50]; fracture incidence [37,48,49,51,53]; and bone turnover markers (BTMs) [27,34,35,36,40,46,58]. Other studies evaluated lean mass or body composition as secondary outcomes [27,34,41], and some included vitamin D or calcium measurements [28]. Due to considerable heterogeneity in population characteristics, study duration, and dietary interventions, random-effects meta-analysis was applied only to outcomes reported by three or more comparable studies. Studies reporting fewer or non-comparable endpoints were summarised narratively. A detailed summary of the included studies is presented in Table 2.

### 3.3. Quality Assessment

A concise summary of the risk of bias (ROB) for all included RCTs is presented in Figure 2. Overall, the evidence base is moderately strong. Selection and performance biases were the primary concerns. Reporting and attrition biases appear minimal, and the most recent studies tend to be of higher methodological quality. The methodological quality of the included non-randomised studies was generally moderate to high based on the JBI Critical Appraisal Checklists (Figure 2A). Most studies clearly defined inclusion criteria, exposures, and outcomes, and used appropriate statistical analyses. However, several cross-sectional analyses showed moderate risk due to limited control for confounding factors. Large prospective cohorts demonstrated consistently low risk of bias across domains. Overall, the evidence base was methodologically robust, with minimal concerns regarding internal validity (Figure 2B, Appendix A).

### 3.4. Effects of Different Dietary Patterns on Femoral Neck Bone Mineral Density

The meta-analysis using the extracted data presents the pooled standardised mean differences (SMDs) in femoral neck BMD across different dietary pattern categories (Figure 3). A total of 11 studies comprising 12 independent comparisons and 12,191 participants were included in this meta-analysis [28,39,40,42,43,45,46,50,56,57,59]. For the Mediterranean diet subgroup, six studies involving 1674 participants in the intervention arms and 1699 controls were analysed. The pooled estimate indicated no statistically significant difference in femoral neck BMD between groups (SMD = 0.08, 95% CI: –0.08 to 0.23, *p* = 0.26). Between-study heterogeneity was moderate (*I*^2^ = 65%, χ^2^ = 14.36, *p* = 0.01), suggesting variability in study outcomes [28,40,45,50,56,59]. In contrast, the calorie restriction diet subgroup included three trials, with a total of 296 participants in the experimental group and 154 controls. The pooled result demonstrated a non-significant reduction in femoral neck BMD among calorie-restricted participants (SMD = –1.06, 95% CI: –5.16 to 3.05, *p* = 0.27), with high heterogeneity (*I*^2^ = 98%, χ^2^ = 141.05, *p* < 0.00001), indicating marked inconsistency between study results [42,43,46]. For high-protein diet, only one large cross-sectional analysis contributed to this subgroup [57]. The analysis revealed a statistically significant positive effect of high protein intake on femoral neck BMD (SMD = 3.00, 95% CI: 2.94 to 3.06, *p* < 0.00001).

Similarly, the low-carbohydrate diet subgroup, represented by a single RCT [39], showed a modest but statistically significant improvement in femoral neck BMD compared to the control group (SMD = 1.01, 95% CI: 0.03 to 1.99, *p* = 0.04). Overall, when all dietary patterns were pooled, the total effect size across studies demonstrated no significant difference in femoral neck BMD between dietary interventions and controls (SMD = 0.12, 95% CI: –0.80 to 1.04, *p* = 0.77). Heterogeneity across all comparisons was substantial (*I*^2^ = 100%, τ^2^ = 1.86, χ^2^ = 4613.64, *p* < 0.00001), reflecting the variability across dietary patterns and study designs. Subgroup differences were statistically significant (χ^2^ = 1834.19, *df* = 3, *p* < 0.00001, *I*^2^ = 99.8%), confirming that the magnitude and direction of effects differed markedly between diet categories. The funnel plot for femoral neck BMD revealed a symmetrical distribution indicating a low risk of publication bias (Appendix A).

### 3.5. Effects of Different Dietary Patterns on Lumbar Spine Bone Mineral Density

The pooled SMD for lumbar spine BMD across various dietary pattern subgroups was also analysed (Figure 4). A total of 14 comparisons from 13 studies involving 4495 participants (experimental n = 2068; control n = 1951) were included in this analysis [28,38,40,42,43,44,45,46,50,55,56,58,59]. The Mediterranean diet subgroup included seven studies, with a total of 1736 participants in the intervention arms and 1764 in the control groups [28,40,45,50,56,58,59]. The pooled analysis demonstrated a non-significant effect of the Mediterranean diet on lumbar spine BMD (SMD = 0.60, 95% CI: –0.30 to 1.49, *p* = 0.19). Between-study heterogeneity was substantial (*I*^2^ = 99%, τ^2^ = 11.45, χ^2^ = 845.74, *p* < 0.00001), suggesting notable variation across studies. Only one small study of 21 participants investigated ketogenic diet and its effect on lumbar spine BMD (SMD = 0.96, 95% CI: –0.05 to 1.87, *p* = 0.04) [44]. One small trial assessed the high-protein diet and found no significant difference in lumbar spine BMD between intervention and control groups (SMD = 0.31, 95% CI: –0.50 to 1.11, *p* = 0.45) [55]. Four studies examined a calorie restriction diet, including [38,42,43,46]. This subgroup contained a total of 310 participants in the intervention groups and 164 in the control groups. The pooled result showed a non-significant reduction in lumbar spine BMD in calorie restriction groups compared to controls (SMD = –1.52, 95% CI: –4.39 to 1.36, *p* = 0.30). Between-study heterogeneity was high (*I*^2^ = 99%, τ^2^ = 8.50, χ^2^ = 302.01, *p* < 0.00001).

When all dietary patterns were pooled, the overall effect estimate indicated no significant difference in lumbar spine BMD between dietary interventions and control groups (SMD = 0.04, 95% CI: –1.12 to 1.03, *p* = 0.94). Heterogeneity among the included studies was substantial (*I*^2^ = 100%, τ^2^ = 3.84, χ^2^ = 1257.44, *p* < 0.00001). Subgroup analysis revealed no significant differences between dietary categories (χ^2^ = 3.11, *df* = 3, *p* = 0.38, *I*^2^ = 3.4%). The funnel plot constructed for lumbar spine BMD showed a slight asymmetry, suggesting a moderate risk of publication bias (Appendix A).

### 3.6. Effects of Different Dietary Patterns on Total Hip Bone Mineral Density

For total hip BMD evaluation according to dietary pattern subgroups, a total of eight studies were included in this analysis, encompassing 43,873 participants (experimental n = 21,669; control n = 21,204) [34,38,42,43,46,53,56,58] (Figure 5). The Mediterranean diet subgroup comprised three studies with large pooled sample sizes (n = 21,333 experimental; n = 21,013 control). The combined effect estimate indicated no significant difference in total hip BMD between the Mediterranean diet and control groups (SMD = 0.10, 95% CI [–0.22, 0.42], *p* = 0.32). Substantial between-study heterogeneity was observed (τ^2^ = 0.02; χ^2^ = 11.68, *df* = 2, *p* = 0.003; *I*^2^ = 79%) [53,56,58]. The calorie restriction subgroup included five studies (n = 336 experimental; n = 191 control) [34,38,42,43,46]. The pooled analysis showed a non-significant reduction in total hip BMD in calorie restriction groups compared to controls (SMD = –0.22, 95% CI [–0.73, 0.28], *p* = 0.29). Between-study heterogeneity remained high (τ^2^ = 0.12; χ^2^ = 13.87, *df* = 4, *p* = 0.008; *I*^2^ = 71%). When all dietary interventions were combined, the overall pooled estimate demonstrated no significant effect of dietary patterns on total hip BMD (SMD = –0.07, 95% CI [–0.36, 0.21], *p* = 0.57). Considerable overall heterogeneity was present (τ^2^ = 0.09; χ^2^ = 40.82, *df* = 7, *p* < 0.00001; *I*^2^ = 91%). Subgroup differences between Mediterranean and calorie restriction diets were not statistically significant (χ^2^ = 2.64, *df* = 1, *p* = 0.10, *I*^2^ = 62.2%).

### 3.7. Effects of Different Dietary Patterns on Whole-Body Bone Mineral Density

Nine studies comprising 43,829 participants for whole-body BMD across dietary pattern subgroups (experimental n = 22,030; control n = 21,799) were included in this analysis [28,35,38,41,42,50,53,55,56] (Figure 6). The Mediterranean diet subgroup included four comparisons [28,50,53,56] with a combined sample of n = 21,897 (experimental) and n = 21,664 (control). Individual study SMDs were SMD = 0.20, 95% CI [0.07, 0.33] [56], SMD = 0.07, 95% CI [−0.20, 0.34] [50], SMD = 0.00, 95% CI [−0.02, 0.02] [53], and SMD = 0.06, 95% CI [−0.06, 0.17] [28]. The pooled effect for the Mediterranean diet was not statistically significant (SMD = 0.07, 95% CI [−0.07, 0.21]; T = 1.52, *df* = 3, *p* = 0.23). Between-study heterogeneity for this subgroup was moderate to substantial (τ^2^ = 0.01; χ^2^ = 9.47, *df* = 3, *p* = 0.02; *I*^2^ = 68%). Two small trials evaluated calorie restriction diets [38,42] (combined n = 33 experimental, n = 29 control). Individual study SMDs were SMD = −0.44, 95% CI [−1.26, 0.38] [38] and SMD = −0.11, 95% CI [−0.74, 0.53] [42]. The pooled estimate for calorie restriction was non-significant and imprecise (SMD = −0.23, 95% CI [−2.30, 1.84]; T = 1.42, *df* = 1, *p* = 0.39), with low subgroup heterogeneity (τ^2^ = 0.00; χ^2^ = 0.40, *df* = 1, *p* = 0.53; *I*^2^ = 0%).

A single small trial examined a high-protein intervention [55] (n = 12 per arm). That study’s effect estimate was not statistically significant (SMD = −0.10, 95% CI [−0.90, 0.70]; Z = 0.25, *p* = 0.80). Heterogeneity was not applicable for this single-study subgroup. Two trials assessed low-carbohydrate diets [35,41] (combined n = 88 experimental, n = 94 control). Individual SMDs were SMD = −0.12, 95% CI [−0.48, 0.24] [35] and SMD = −0.41, 95% CI [−0.91, 0.08] [41]. The pooled low-carbohydrate estimate was non-significant (SMD = −0.22, 95% CI [−2.01, 1.57]; T = 1.56, *df* = 1, *p* = 0.36), with negligible subgroup heterogeneity (τ^2^ = 0.00; χ^2^ = 0.89, *df* = 1, *p* = 0.35; *I*^2^ = 0%). When all dietary patterns were pooled, the overall effect on whole-body BMD was small and not statistically significant (SMD = 0.03, 95% CI [−0.07, 0.14]; T = 0.70, *df* = 8, *p* = 0.50). Overall, between-study heterogeneity was moderate (τ^2^ = 0.01; χ^2^ = 13.95, *df* = 8, *p* = 0.08; *I*^2^ = 46%). The test for subgroup differences did not reach statistical significance (χ^2^ = 6.51, *df* = 3, *p* = 0.09; *I*^2^ = 53.9%).

### 3.8. Effect of Mediterranean Diet on Fracture Risk

The pooled hazard ratios (HRs) for fracture outcomes comparing Mediterranean dietary patterns with control diets were also analysed (Figure 7). Fifteen comparisons from the included studies contributed to four fracture subgroups: any fracture [28,37,49,51,53], hip fracture [47,48,51,52,53,54,60,61], vertebral fracture, and wrist fracture [51].

For any fracture, the pooled estimate showed a statistically significant lower hazard in participants following a Mediterranean pattern compared with controls (HR = 0.95, 95% CI [0.93, 0.97]; Z = 5.23, *p* < 0.00001). Between-study heterogeneity for this subgroup was substantial (Chi^2^ = 13.12, *df* = 4, *p* = 0.01; *I*^2^ = 70%) [28,37,49,51,53]. The pooled HR for hip fracture also indicated a statistically significant reduction in hazard with Mediterranean patterns (HR = 0.93, 95% CI [0.90, 0.96]; Z = 4.11, *p* < 0.0001). Heterogeneity in this subgroup was high (Chi^2^ = 79.47, *df* = 7, *p* < 0.00001; *I*^2^ = 91%) [47,48,51,52,53,54,60,61]. The vertebral fracture subgroup (single comparison) showed a non-significant pooled hazard (HR = 1.06, 95% CI [0.87, 1.29]; Z = 0.58, *p* = 0.56); heterogeneity was not applicable for this single study [51].

When all fracture outcomes were pooled across subgroups, the overall hazard ratio favoured the Mediterranean diet (HR = 0.95, 95% CI [0.93, 0.96]; Z = 6.35, *p* < 0.00001). The overall heterogeneity across all comparisons was high (Chi^2^ = 98.21, *df* = 14, *p* < 0.00001; *I*^2^ = 86%). The test for differences between the predefined fracture subgroups did not reach statistical significance (Chi^2^ = 5.63, *df* = 3, *p* = 0.13; *I*^2^ = 46.7%). The funnel plot for fracture risk no marked evidence of asymmetry, suggesting a low publication bias (Appendix A).

### 3.9. Effect of Different Diets on Bone Turnover Markers

Seven included studies assessed circulating markers of bone turnover or related regulators across a range of dietary interventions. The markers reported across studies included established resorption markers (serum CTX, urinary N-telopeptide [uNTx], TRAP5b), formation markers (bone-specific alkaline phosphatase (BAP, P1NP, osteocalcin), and regulatory proteins (sclerostin, osteoprotegerin [OPG], PTH, DKK1, FGF-23), as well as a broad panel of inflammatory cytokines in one cohort (Table 3).

Brinkworth et al. [35] reported longitudinal increases in a resorption marker, serum β-CrossLaps, over 12 months in both diet arms: the very-low-carbohydrate (LC) group rose from 0.44 ± 0.20 to 0.54 ± 0.23 µg/L (+24%), and the low-fat (LF) group from 0.47 ± 0.23 to 0.62 ± 0.26 µg/L (+32%); there was no statistically significant difference between diets. Carter et al. [36] likewise found no significant between-group differences in bone turnover: mean uNTx and BAP changed modestly over 1–3 months in both low-carbohydrate dieters and controls, with non-significant differences. Calorie restriction (CR) was associated with increases in bone resorption markers and decreases in some formation markers in the long-term RCT by Villareal et al. [46]. Compared with ad libitum controls, the CR group exhibited significant rises in serum CTX and TRAP5b as early as 6 months that remained elevated at 12 months; BAP decreased significantly at 12 and 24 months, while P1NP did not change significantly. In a lifestyle trial in frail obese older adults, the diet-only arm (without exercise) showed increases in CTX (+31%), osteocalcin (OCN, +24%), and P1NP (+9%) and an increase in sclerostin (+10.5%), whereas the exercise arm showed decreases in CTX (−13%), OCN (−15%), and P1NP (−15%) and prevented the rise in sclerostin; combined diet + exercise and control arms showed no significant changes [34]. Carter et al. [36] also reported weight loss in the low-carbohydrate dieters (6.39 kg vs. 1.05 kg in controls at 3 months), with no significant between-group differences in the reported turnover markers.

In the 2-year walnut supplementation sub-study [40], a broad panel of bone-related proteins was measured (ACTH, DKK1, OPG, osteocalcin, osteopontin, sclerostin, PTH, FGF-23). No significant differences were observed between walnut and control groups for these biomarkers; the authors noted a non-significant trend toward higher PTH in the walnut group (*p* = 0.054). Cervo et al. [27] examined 24 circulating cytokines in relation to Mediterranean-pattern adherence (MEDI-LITE) and reported largely null associations: IL-7 showed an inverse crude association with MEDI-LITE that did not remain significant after multiplicity correction, and no other cytokines were significantly associated with the diet score. MEDI-LITE adherence was not associated with BMD at any site in this cohort; a small positive association was observed between MEDI-LITE and appendicular lean mass (ALMBMI). The investigators reported that IL-7 mediated only ~10% of the observed diet–falls association.

Although not a bone turnover biomarker study per se, Moradi et al. [58] examined dietary patterns stratified by the TGF-β1 T869→C polymorphism and reported that higher Mediterranean diet adherence was associated with higher lumbar spine Z-scores and reductions in BMI, fat mass, and visceral fat (*p* < 0.05), whereas traditional diet patterns in certain genotypes were associated with lower lumbar spine Z-scores. These findings relate to bone status and body composition measures that are mechanistically relevant to bone turnover but do not directly report classical BTMs.

### 3.10. Effect of Different Diets on Vitamin D and Calcium Status

A total of six studies assessed the influence of dietary interventions—primarily Mediterranean and calorie restriction diets on circulating vitamin D and calcium levels (Table 4). The results demonstrated variable yet modest changes across studies. In a randomised Mediterranean diet intervention, Jennings et al. [28] reported higher mean serum 25-hydroxyvitamin D [25(OH)D] concentrations in the intervention group (5.2 ng/mL; 95% CI 1.7–8.8) compared with the control group (3.8 ng/mL; 95% CI 0.7–6.9). A large prospective analysis by Byberg et al. [49] found nearly identical mean vitamin D intakes between Mediterranean diet adherents (5.56 mg/day) and controls (5.58 mg/day), indicating minimal difference in dietary vitamin D intake.

Among calorie restriction trials, Hinton et al. [38] observed slightly higher mean serum vitamin D levels in the calorie restriction (CR) group (6 ± 7.5 ng/mL) compared with controls (5.5 ± 6.8 ng/mL). In a longer intervention, Villareal et al. [46] reported an increase in serum 25(OH)D from baseline in the CR group (mean + 1.9 ng/mL ± 0.7; post-intervention 29.6 ± 0.6 ng/mL) relative to a marginal decline in the ad libitum (AL) group (−0.3 ± 0.9; post-intervention 29.3 ± 0.9 ng/mL). Similarly, Pop et al. [42] reported comparable 25(OH)D concentrations between groups (68.0 ± 24.2 nmol/L vs. 65.9 ± 17.8 nmol/L). The data on calcium intake followed a comparable trend. Hinton et al. [38] noted slightly higher calcium intake in the CR group (20 ± 210 mg/day) compared with controls (15 ± 200 mg/day), while Byberg et al. [49] observed marginally higher calcium intake among Mediterranean diet adherents (1298 mg/day) relative to controls (1254 mg/day).

Overall, across Mediterranean and calorie restriction dietary patterns, changes in vitamin D status and calcium intake were small and not markedly different between intervention and control groups [28,38,42,46,49].

## 4. Discussion

This systematic review and meta-analysis synthesised evidence from 30 studies (RCTs, prospective cohorts, cross-sectional analyses, and controlled feeding studies) that examined the effects of major dietary patterns, principally Mediterranean-style diets, calorie restriction, low-carbohydrate/ketogenic diets, and higher-protein diets on skeletal outcomes including BMD, fracture incidence, bone turnover markers, and vitamin D/calcium status. Overall, pooled estimates indicated no consistent, clinically important effect of dietary patterns on BMD at the femoral neck, lumbar spine, total hip, or whole body when all diets were combined, although subgroup analyses revealed important pattern-specific heterogeneity (e.g., smaller pooled effects for Mediterranean diets vs. more variable effects for CR and protein-rich interventions). Mediterranean dietary patterns were associated with modest reductions in fracture hazard (pooled HR ≈ 0.93–0.95 for hip/any fracture), while CR regimens frequently produced increases in bone resorption markers and reductions in some formation markers, even when absolute changes in BMD were inconsistent across trials. Changes in circulating 25(OH)D and calcium intake were small and inconsistent between intervention and control arms across studies.

The finding that adherence to a Mediterranean dietary pattern is associated with a lower hazard of fractures (hip and any fracture) aligns with prior observational syntheses showing an inverse relation between Mediterranean-style diets and fracture risk [62,63] and with a recent meta-analysis indicating a dose–response protective association for hip fracture [64]. The plausible mechanisms include higher intakes of fruits, vegetables, whole grains, legumes, fish, and olive oil that collectively supply calcium, magnesium, vitamin K, potassium, polyphenols, and anti-inflammatory constituents supportive of bone remodelling and microarchitecture [54]. Despite the protective signal for fractures, pooled effects for BMD across Mediterranean-diet studies were generally null or small and heterogeneous, suggesting that fracture reduction may be mediated by pathways other than large changes in DXA-measured BMD alone (e.g., improved bone quality, reduced falls via better muscle mass/function, or reduced inflammation) [27,28]. Notably, the Concord cohort found MEDI-LITE scores associated with small increases in appendicular lean mass and lower fall risk but no cross-sectional BMD benefit, and the mediation analysis suggested only a modest cytokine (IL-7) contribution to the falls effect, consistent with the concept that fracture risk reduction may operate through multi-factorial mechanisms beyond areal BMD [27].

In contrast, calorie restriction diets produced a clearer and concordant effect on bone remodeling markers. Multiple trials reported increases in bone resorption markers (serum CTX, TRAP5b) and decreases in bone formation markers (e.g., BAP) or net suppression of bone formation over medium-term follow-up (6–24 months). A two-year RCT of 25% CR demonstrated significant rises in CTX and TRAP5b by 6–12 months and sustained declines in BAP at 12–24 months [46]. These biochemical changes are consistent with earlier experimental work showing that weight loss induced by negative energy balance preferentially increases bone turnover and may lead to site-specific BMD loss unless counteracted by mechanical loading or resistance exercise [46,65]. Thus, although some CR trials reported minimal net BMD change (depending on duration, degree of weight loss, and concomitant exercise), the pattern of elevated resorption is biologically plausible and clinically relevant because prolonged increases in resorption can translate into measurable BMD loss and increased fragility over time. Evidence for low-carbohydrate and ketogenic diets was mixed and limited by small sample sizes and short follow-ups. Some controlled feeding and trial data suggested modest increases in femoral neck BMD or neutral effects on spine BMD, but BTMs in certain short-term KD studies have pointed to transient increases in bone resorption or suppressed formation [35,44]. Recent systematic reviews specifically addressing KD and bone have reported that evidence remains inconclusive: many KD studies are small, heterogeneous in macronutrient composition and duration, and often study athletes or people with epilepsy rather than older adults at fracture risk [66]. In addition, since this diet is frequently followed by Type 2 diabetic patients [67], the effects should be considered alongside anti-hyperglycaemic agents, which also affect bone outcomes [68].

Higher-protein diets showed a generally favourable association with hip and femoral BMD in observational datasets and umbrella reviews, and one large cross-sectional analysis [57] contributed a notable positive SMD for femoral neck BMD in our pooled analysis (albeit cross-sectional data limit causal inference). Meta-analyses and umbrella reviews support a protective role of protein above the RDA for hip fracture and BMD maintenance in older adults, likely mediated through preservation of lean mass, stimulation of IGF-1, and improved muscle–bone mechanical coupling [69,70]. Furthermore, when examining protein diets’ effects, the protein intake and the quality of the protein ingested (level of indispensable amino acids) are important parameters that should be considered. Vitamin D and calcium status changed little across most dietary interventions in the included trials. Trials that measured 25(OH)D reported small, inconsistent differences [28,46], and large cohort dietary assessments found comparable vitamin D or calcium intakes between higher and lower Mediterranean adherence strata [49]. This suggests that the skeletal effects attributed to whole dietary patterns are unlikely to be fully explained by differences in dietary vitamin D or total calcium intake alone in the studies reviewed.

The heterogenous pattern of findings across dietary regimes highlights several mechanistic pathways. Mediterranean diets sully a matrix of micronutrients (vitamin K, Mg, vitamin C), unsaturated fatty acids, polyphenols, and antioxidants that can reduce oxidative stress and low-grade inflammation—factors implicated in osteoclast activation and bone loss [37,54]. Conversely, diets that heavily restrict energy (CR) or omit major food groups (very low energy KD with limited fruit/whole-grain intake) can reduce mechanical loading (via weight loss), alter endocrine drivers of bone remodeling (leptin, IGF-1), and change calcium or vitamin D bioavailability, thereby increasing resorption [46,71]. Higher dietary protein may support bone through anabolic effects on muscle and bone matrix and by improving calcium economy when accompanied by adequate calcium intake, although extremely high sustained protein without sufficient alkalinising foods might increase acid load and have adverse effects in vulnerable subgroups [72].

The strengths of this review include a comprehensive search strategy, inclusion of diverse study designs, and focused outcomes that are clinically relevant (site-specific BMD, fracture, BTMs, vitamin D/calcium). A subgroup analysis was performed by diet pattern and prioritised random-effects models where appropriate to account for between-study heterogeneity. However, some limitations temper confidence in conclusion. First, heterogeneity was substantial for many pooled outcomes (*I*^2^ frequently > 70–90%), reflecting variation in study populations (young vs. older adults; healthy vs. metabolic syndrome), intervention definitions (heterogeneity within “Mediterranean” or “ketogenic” labels), comparator arms, adherence measurement, follow-up duration, and outcome measurement methods (different DXA sites, biomarker assays). Second, many RCTs were small and underpowered for BMD or fracture endpoints; conversely, large cohort studies provide power but are observational and susceptible to residual confounding and exposure misclassification. Third, biochemical responses (BTMs) were assessed in only a subset of trials and often at different timepoints; while BTMs are sensitive to short-term changes, their translation to long-term fracture risk depends on magnitude and duration of change. Fourth, several trials combined diet with exercise or behavioural support (e.g., PREDIMED-Plus secondary analyses), making it difficult to isolate the independent effect of diet. Finally, measurement of vitamin D and calcium differed (dietary intake vs. serum 25[OH]D), complicating synthesis.

The body of evidence supports recommending Mediterranean-style dietary patterns as part of holistic strategies to reduce fracture risk in older adults, given consistent observational signals for lower hip or overall fractures and broader cardiometabolic benefits of this pattern [37,47]. Clinicians and dietitians should, however, be cautious when endorsing prolonged energy-restricted diets for otherwise healthy non-obese adults without concurrent bone-protective measures such as resistance exercise, adequate calcium and vitamin D, and monitoring, because CR is repeatedly associated with increases in bone resorption markers and potential site-specific BMD loss. As per Villareal et al. [46], prolonged CR produced biochemical and DXA changes consistent with increased bone turnover and BMD decline. For low-carbohydrate and ketogenic regimens, current evidence is insufficient and heterogenous; clinicians should monitor bone health in individuals on long-term KD, especially older adults, until larger, longer-term trials clarify net skeletal effects [67]. High-protein diets appear compatible with bone health and may confer benefits for hip BMD and fracture reduction when calcium intake is adequate and exercise preserves lean mass; however, evidence from RCTs with fracture endpoints remains sparse [69].

High-quality, adequately powered RCTs are urgently required to advance understanding of the relationship between dietary patterns, calorie restriction, and bone health. Future studies should employ standardised and clearly defined dietary interventions, accompanied by objective measures of adherence. Older adults at increased risk of fracture should be prioritised as the primary target population. Methodologically, such trials should integrate DXA with complementary bone quality assessments, such as high-resolution peripheral quantitative compute tomography (HR-pQCT) and finite element analysis, alongside serial measurements of BTMs at prespecified timepoints. Clinically meaningful endpoints, including incident fractures and fall, should be evaluated concurrently with mechanistic mediators such as changes in muscle mass and strength, inflammatory biomarkers, and vitamin D/calcium status. Of particular value would be trials directly comparing CR combined with resistance exercise versus CR alone, given the evidence suggesting that resistance exercise may attenuate CR-induced bone loss [34,66]. Additionally, longer-term follow-up of ketogenic diet (KD) interventions in older populations is warranted, with close monitoring to ensure nutrient adequacy and safety. Finally, the establishment of harmonised reporting standards, for example, consistent definitions of dietary interventions, standardised reporting of BMD in both g/cm^2^ and T/Z-scores, and uniform presentation of BTMs, would substantially enhance data comparability across studies. Given the methodological variability and heterogeneity across existing studies, clinical recommendations should emphasise overall dietary quality, favouring Mediterranean-style eating, alongside adequate calcium and vitamin D intake, and structured resistance training, particularly during weight-loss interventions. Future rigorously designed RCTs employing standardised dietary protocols and clinically relevant skeletal outcomes are essential to establish causal relationships and refine dietary guidelines for optimal bone health.

## 5. Conclusions

Overall, current evidence indicates that dietary patterns exert diverse and complex influence on skeletal health. Cohort and clinical trial data consistently demonstrate a protective association between adherence to a Mediterranean dietary pattern and reduced fracture risk. In contrast, CR has been shown to elevate biochemical markers of bone resorption and, over time, may suppress bone formation. Evidence regarding ketogenic and high-protein diets remains inconclusive; however, higher protein intake appears to support BMD when accompanied by sufficient calcium intake and regular resistance exercise.

## Figures and Tables

**Figure 1 nutrients-17-03845-f001:**
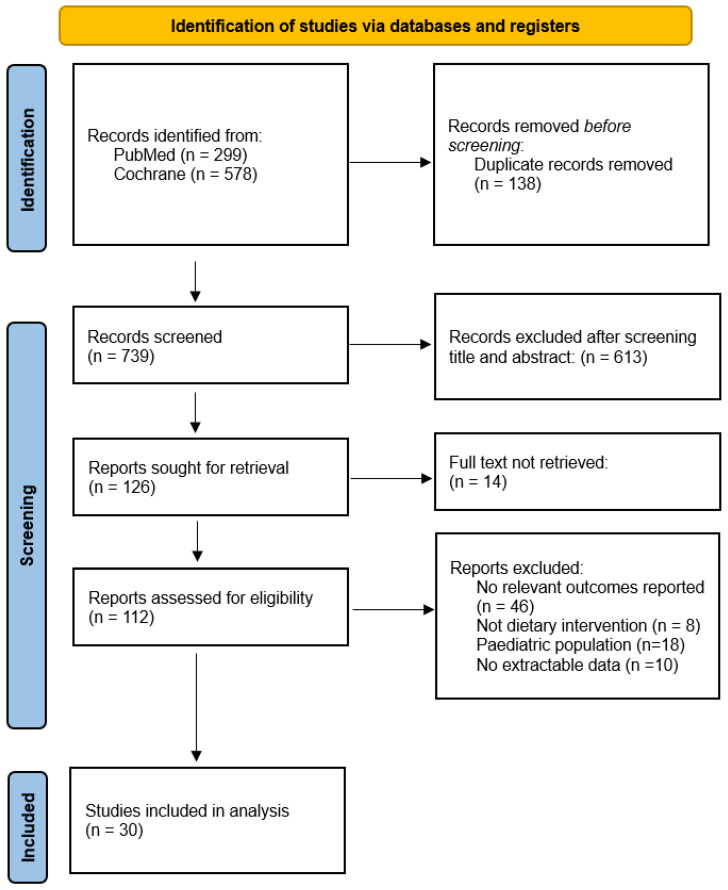
PRISMA flowchart of study search and selection process.

**Figure 2 nutrients-17-03845-f002:**
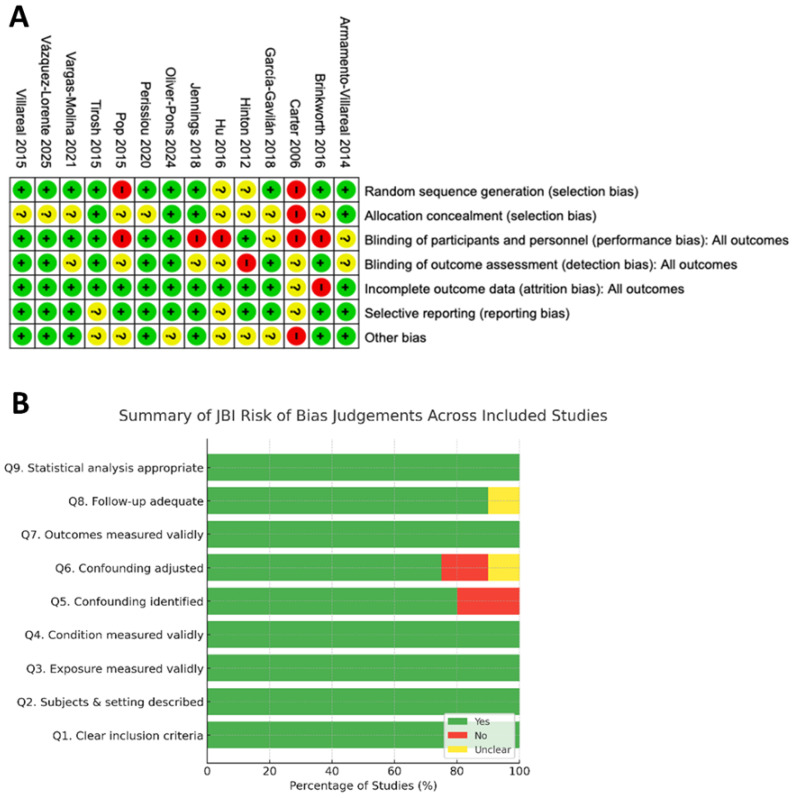
Assessment of risk of bias for the included RCTs [28,34,35,36,37,38,39,40,41,42,43,44,45,46] (**A**), and summary of methodological quality based on JBI Critical Appraisal Checklists. Proportion of “Yes” (green), “No” (red), and “Unclear” (yellow) responses across nine domains of the Joanna Briggs Institute (JBI) critical appraisal tool applied to the included non-randomised studies [27,47,48,49,50,51,52,53,54,55,56,57,58,59,60,61] (**B**).

**Figure 3 nutrients-17-03845-f003:**
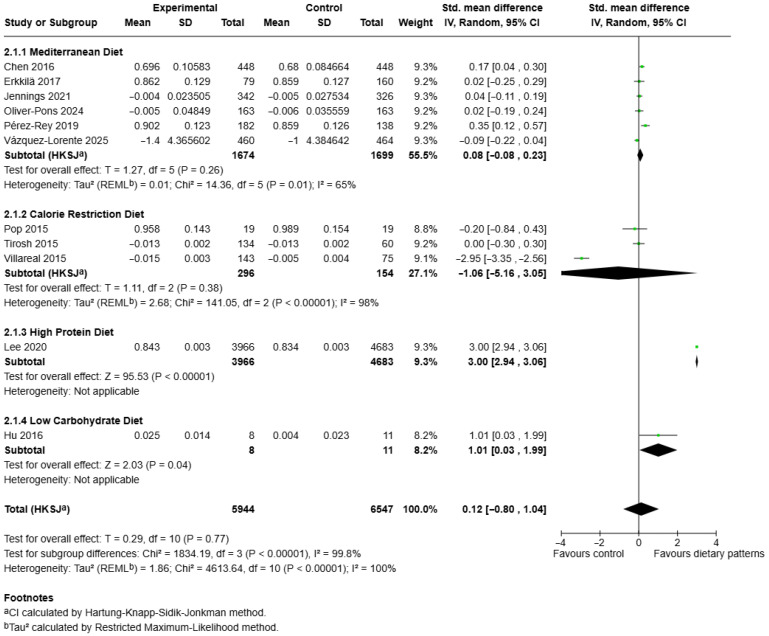
Forest plot of the effect of different dietary patterns on femoral neck BMD. Standardised mean differences in subgroups and total effect are presented with 95% confidence intervals using the random effects model [28,39,40,42,43,45,46,50,56,57,59].

**Figure 4 nutrients-17-03845-f004:**
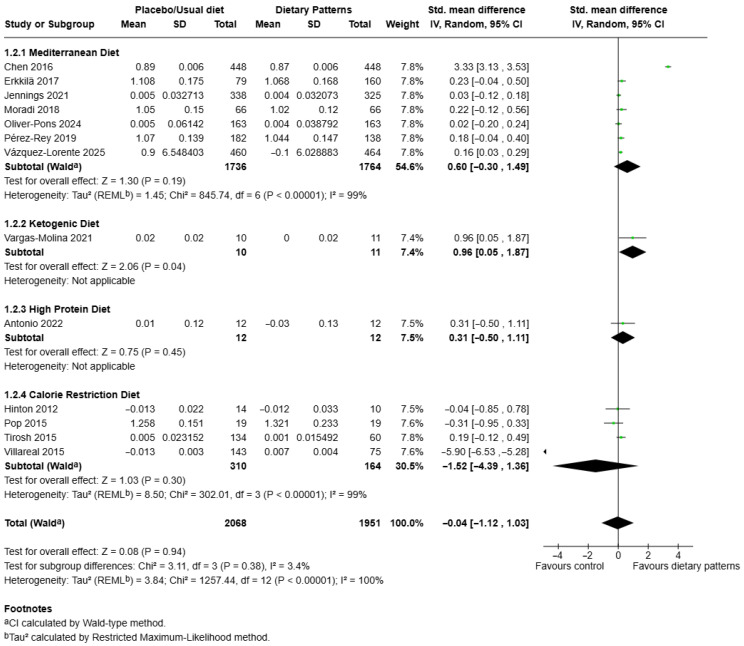
Forest plot showing the effect of different dietary patterns on lumbar spine BMD. Standardised mean differences in subgroups and total effect are presented with 95% confidence intervals using the random effects model [28,38,40,42,43,44,45,46,50,55,56,58,59].

**Figure 5 nutrients-17-03845-f005:**
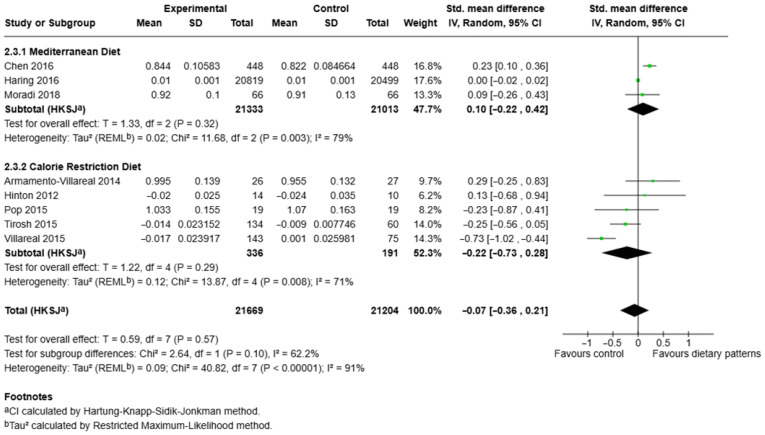
Forest plot showing the effect of different dietary patterns on total hip BMD. Standardised mean differences in subgroups and total effect are presented with 95% confidence intervals using the random effects model [34,38,42,43,46,53,56,58].

**Figure 6 nutrients-17-03845-f006:**
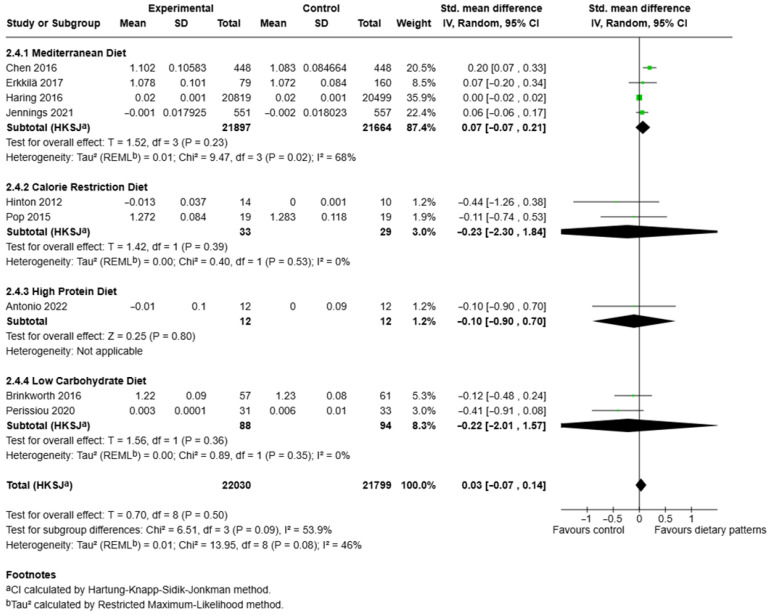
Forest plot showing the effect of different dietary patterns on whole-body BMD. Standardised mean differences in subgroups and total effect are presented with 95% confidence intervals using the random effects model [28,35,38,41,42,50,53,55,56].

**Figure 7 nutrients-17-03845-f007:**
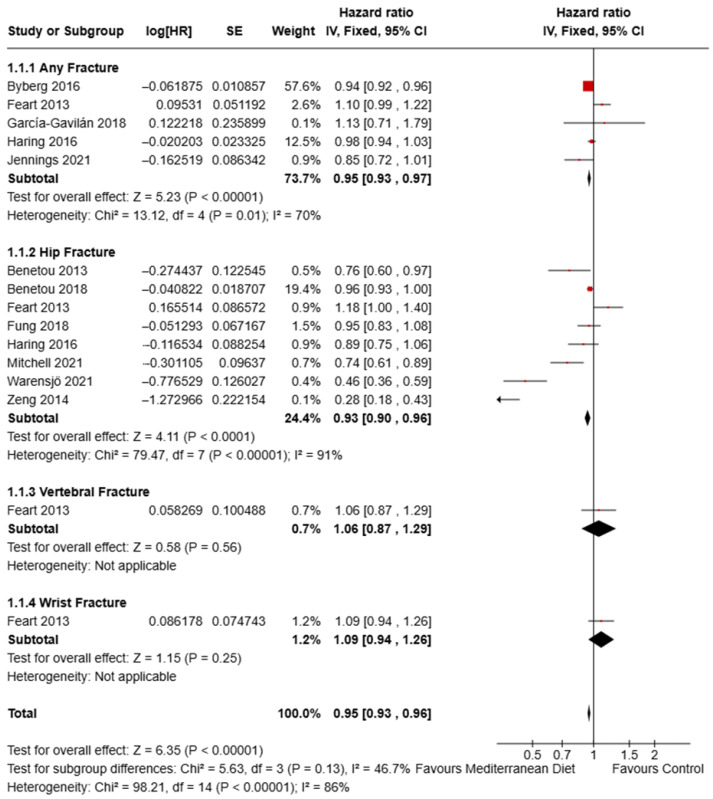
Forest plot showing the effect of Mediterranean dietary patterns on fracture risk (hazard ratios) [28,37,47,48,49,51,52,53,54,60,61].

**Table 1 nutrients-17-03845-t001:** PICOS criteria for inclusion of studies.

Parameter	Description	Search Terms Used
**Population**	Adults (>18 y), both sexes, with no restriction on baseline musculoskeletal status	osteoporosis OR bone mineral density OR fractures OR skeletal health OR musculoskeletal disorders
**Intervention**	Dietary patterns (Mediterranean diet, ketogenic diet, high-protein diet, calorie restriction, low-carb diets)	“Mediterranean diet” OR “ketogenic diet” OR “low carbohydrate diet” OR “protein-rich diet” OR “calorie restriction” OR “dietary pattern” OR “dietary intervention”
**Comparator**	Regular diet, placebo, or other dietary interventions	control OR usual care OR standard diet
**Outcomes**	Bone mineral density (femoral neck, lumbar spine, total hip), fracture incidence, falls incidence, bone turnover markers (CTX, P1NP), calcium/vitamin D status (serum 25(OH)D, calcium intake/balance), lean body mass	BMD OR bone density OR fractures OR falls OR CTX OR P1NP OR vitamin D OR calcium OR lean mass OR body composition
**Study Design**	Randomised controlled trial (RCT) or prospective cohort study	randomised OR trial OR cohort OR longitudinal study

**Table 2 nutrients-17-03845-t002:** Characteristics of the included studies.

**First Author (Year)**	**Country**	**Study Design**	**N**	**Age** **(Mean ± SD/Range) in Years**	**Sex** **(% M/F)**	**Health Status**	**Diet Type/Intervention**	**Comparator**	**Duration/Follow-Up**	**Key Outcomes**
Antonio et al. (2018) [55]	USA (JISSN)	Cross-sectional/intervention	24	Young trained adults	0/100	Resistance-trained women	High-protein diet	Habitual diet	6 months	Bone markers/BMD.
Armamento-Villareal et al. (2014) [34]	USA	RCT/lifestyle intervention	107	69 ± 4	Mixed	Frail, obese older adults	Lifestyle therapy (weight loss + exercise)	Usual care/baseline	52 weeks	Thigh muscle volume ↔ BMD (DXA); lean mass–BMD correlation.
Benetou et al. (2018) [47]	Europe (multi-country)	Prospective cohort (CHANCES)	140,775	Older adults	Mixed (116,176 women, 24,599 men)	Community-dwelling older adults	Mediterranean diet adherence (mMED)	Lower adherence	Variable (cohort-dependent)	Incident hip fracture ↓ with higher MD adherence.
Benetou et al. (2013) [48]	Europe (EPIC, 8 countries)	Prospective cohort	188,795	48.6 ± 10.8	48,814 M/139,981 F	General population	Mediterranean diet score (10-point)	Lower adherence	9 years	Incident hip fractures (protective association).
Brinkworth et al. (2016) [35]	Australia	Randomised cross-over RCT	118	51.3 ± 7.1 (24–64)	Mixed	Abdominal obesity ± metabolic risk	Very-low-carbohydrate, energy-restricted diet	Isocaloric low-fat diet	12 months	Total-body BMC/BMD (DXA); BTMs NR.
Byberg et al. (2016) [49]	Sweden	Prospective cohort (pooled)	71,333	~60	53/47	Adults free of CVD/cancer	Mediterranean diet (mMED)	Low adherence	15 years	Incident hip fracture (n = 3175).
Carter et al. (2006) [36]	USA	RCT	30	40.08 ± 6.0	Mixed	Adults on low-carbohydrate diet	Low-carbohydrate diet	Control diet	1–3 months	Bone turnover markers (uNTx, BAP); no significant difference.
Cervo et al. (2021) [27]	Australia (CHAMP)	Prospective cohort	794 (x-sec); 616 (long.)	81.1 ± 4.5	100 M	Older men	Mediterranean diet (MEDI-LITE)	Low adherence	3 years	BMD ↑ with higher score; ALM and falls improved.
Chen et al. (2016) [56]	China	Cross-sectional	2371	40–75	Mixed	Community adults	Mediterranean diet adherence	–	3 years	Higher adherence → higher BMD; BTMs/Vit D NR.
Erkkilä et al. (2017) [50]	Finland	Prospective cohort	554 (F)	67.9 ± 1.9	100 F	Elderly women	Mediterranean and Baltic Sea diets	Lower adherence	3 years	Lumbar, femoral, and total BMD (DXA).
Feart et al. (2013) [51]	France	Prospective cohort	1482	≥67	Mixed	Older community dwellers	Mediterranean diet adherence	Lower adherence	8 years	Incidence of hip, vertebral, wrist fractures.
Fung et al. (2018) [52]	USA	Prospective cohort (NHS and HPFS)	111,048	≥50	Mixed	Postmenopausal women and older men	High diet quality scores (AHEI, aMed, DASH, HEI)	Low diet quality scores	26–32 years	No significant association between diet quality and hip fracture risk.
García-Gavilán et al. (2018) [37]	Spain (PREDIMED)	RCT	870 (subset)	55–80	Mixed	High CVD risk	MedDiet + EVOO/nuts	Low-fat diet	8.9 years	Osteoporotic fracture risk ↓ (EVOO group).
Haring et al. (2016) [53]	USA (WHI-OS)	Prospective cohort (post hoc)	90,014	63.6 ± 7.4 (50–79)	0/100	Postmenopausal women	aMED, HEI-2010, AHEI-2010, DASH	Low adherence	15.9 years	Hip and total fractures; BMD (hip/whole-body); lean mass.
Hinton et al. (2012) [38]	USA	RCT (weight loss)	40	39 ± 1	100% female	Obese women	Calorie restriction diet	Partial regain ± aerobic exercise	12 months	BMD and BTMs after weight loss/regain.
Hu et al. (2016) [39]	USA	RCT	21 (9/12)	52.7 (10.7)/51.8 (11.7)	0/100	Obese women (no DM, CVD, CKD)	Low-carb (<40 g day^−1^)	Low-fat (<30% kcal fat)	12 months	BMD (L1–4, femur, neck T/Z-scores).
Jennings et al. (2018) [28]	Europe (5 countries)	RCT	1142	70.9 ± 4.0	44/56	Older Europeans (osteoporosis subset)	Med-like diet + Vit D_3_ (10 µg day^−1^)	Control diet	1 years	BMD (femoral neck, lumbar, whole-body) ↑; slower bone loss.
Lee et al. (2020) [57]	USA (NHANES)	Cross-sectional analysis	12,812	46.25 ± 0.34	Mixed	With/without CKD	Protein intake levels	–	NR	Protein intake correlated with BMD in CKD and non-CKD groups.
Mitchell et al. (2020) [54]	Sweden	Prospective cohort (COSM and SMC)	50,755	Middle–older adults	Mixed	Healthy adults at baseline	Mediterranean diet (mMED; low–high adherence)	Low mMED adherence	17 years	High adherence reduced hip fracture risk (OR 0.73–0.82); not mediated by BMI or T2DM.
Moradi et al. (2018) [58]	Iran	Cross-sectional	254	57.8 ± 6.1 (46–78)	0/100	Postmenopausal women	Mediterranean/traditional/unhealthy diets	–	NR	BMD (L2–L4, femur); TGF-β_1_ gene–diet interaction.
Oliver-Pons et al. (2024) [40]	Spain (WAHA)	RCT (two-centre)	352 (BMD); 211 subset	63–79	Mixed	Healthy older adults	Walnut-enriched diet (~15% energy)	Usual diet (no walnuts)	2 years	BMD (spine, neck); bone biomarkers (OCN, OPG, sclerostin).
Pérez-Rey et al. (2019) [59]	Spain	Cross-sectional	442	42.7 ± 6.7	0/100	Premenopausal women	Mediterranean diet adherence	–	NR	BMD ↑ with higher MD adherence.
Perissiou et al. (2020) [41]	Australia	RCT (exercise + diet)	64 (33/31)	35.3 ± 9	Mixed	Obese adults (BMI 30–35)	Low-carb (≤50 g day^−1^) + exercise	Standard diet + exercise	8 weeks	Total BMD ↔ lean mass ↓ in LC group.
Pop et al. (2015) [42]	USA	RCT (weight-loss men)	38	58 ± 6	100 M	Overweight/obese men	Moderate weight-loss intervention	–	6 months	Bone quality preserved; BMD/BTMs reported.
Tirosh et al. (2015) [43]	USA (POUNDS LOST)	RCT	424	51.8 ± 8.9 (30–70)	43/57	Overweight/obese adults	Weight-loss diets (HP, HF, HC)	Other diet arms	24 months	BMD (spine, hip, neck); body composition reported.
Vargas-Molina et al. (2021) [44]	Spain	RCT	21 (10/11)	Young adults (resistance-trained)	0/100	Healthy trained women	Ketogenic diet + resistance training	Non-ketogenic diet + training	8 weeks (+3 weeks familiarisation)	BMD/BMC (DXA) slight ↑ in KD; muscle outcomes secondary.
Vázquez-Lorente et al. (2025) [45]	Spain (PREDIMED-Plus)	RCT (secondary analysis)	924 (460/464)	65.1 ± 5.0 (55–75)	51/49	Older adults with metabolic syndrome	Energy-reduced MedDiet + PA + behavioural support	Ad libitum MedDiet (no PA)	3 years	Lumbar spine BMD protective (esp. women); femur no effect.
Villareal et al. (2015) [46]	USA	RCT (caloric restriction)	218	Non-obese younger adults (20–50)	Mixed	Healthy non-obese adults	Caloric restriction (~25%)	Usual diet	2 years	Bone metabolism and BMD changes (DXA).
Warensjö et al. (2021) [61]	Sweden	Longitudinal cohort	82,092	Middle–older adults	Mixed	General population	Combined dietary calcium intake + Mediterranean-style diet (mMED)	Low Ca or low mMED adherence	20 years	Lowest hip fracture risk with Ca ≥800 mg/day + high mMED; risk ↑ with low Ca or mMED (HR 1.50–1.54).
Zeng et al. (2014) [60]	China	Case–control	1452 (726/726)	55–80	Mixed	Elderly urban adults (hip fracture vs. controls)	Diet-quality scores (HEI-2005, aHEI, DQI-I, aMed)	Lower diet quality	NR	Hip fracture risk ↓ (OR ≈ 0.2–0.3 highest vs. lowest quartile).

**Abbreviations:** BMD—bone mineral density; BMC—bone mineral content; BTMs—bone turnover markers; Vit D—vitamin D; Ca—calcium; DXA—Dual-energy X-ray Absorptiometry; ALM—appendicular lean mass; CVD—Cardiovascular Disease; CKD—Chronic Kidney Disease; HP—high-protein; HF—high-fat; HC—high-carbohydrate; PA—physical activity; mMED—Modified Mediterranean Diet Score; EVOO—Extra Virgin Olive Oil; OCN—Osteocalcin; OPG—Osteoprotegerin; NR—not reported.

**Table 3 nutrients-17-03845-t003:** Summary of studies assessing the effect of different diets on bone turnover markers.

Study (First Author, Year)	Design/Population	Dietary Intervention/Comparator	Duration	Biomarkers Assessed	Main Findings
Brinkworth et al., 2016 [35]	RCT, overweight adults	Very-low-carbohydrate (LC) vs. low-fat (LF) diet	12 months	Serum β-CrossLaps	Both LC (+24%) and LF (+32%) ↑ β-CrossLaps; no significant difference between diets.
Carter et al., 2006 [36]	RCT, overweight adults	Low-carbohydrate diet vs. control	3 months	uNTx, BAP, Bone turnover ratio	No significant differences in uNTx or BAP; minor non-significant changes; greater weight loss in low-carb group.
Villareal et al., 2015 [46]	RCT, older adults	Caloric restriction (CR) vs. ad libitum (AL)	24 months	CTX, TRAP5b, BAP, P1NP	CR ↑ CTX and TRAP5b (6–12 mo); ↓ BAP at 12–24 mo; P1NP unchanged.
Armamento-Villareal et al., 2014 [34]	RCT, obese older adults	Diet, exercise, diet + exercise vs. control	12 months	CTX, OCN, P1NP, Sclerostin, IGF-1	Diet ↑ CTX (+31%), OCN (+24%), P1NP (+9%), Sclerostin (+10.5%); Exercise ↓ CTX (−13%), OCN (−15%), P1NP (−15%); combined intervention prevented sclerostin rise.
Oliver-Pons et al., 2024 [40]	RCT, adults	Walnut supplementation vs. control	24 months	ACTH, DKK1, OPG, OCN, OPN, SOST, PTH, FGF-23	No significant group differences; trend toward ↑ PTH (*p* = 0.054).
Cervo et al., 2021 [27]	Prospective cohort	High vs. low Mediterranean diet adherence (MEDI-LITE)	3 years	24 cytokines, BMD, lean mass	No significant cytokine associations after correction; weak inverse link of IL-7 with diet; no BMD associations.
Moradi et al., 2018 [58]	Cross-sectional	Mediterranean, traditional, unhealthy dietary patterns (by TGF-β1 genotype)	—	Lumbar spine Z-score, BMD, body composition	Mediterranean diet ↑ lumbar spine Z-score & ↓ fat measures; traditional diet in C allele carriers ↓ lumbar spine Z-score; no direct BTM data.

**Abbreviations:** uNTX = urinary N-telopeptide of Collagen Type I; BAP = Bone-specific Alkaline Phosphatase; CTX = C-telopeptide of Collagen Type I; TRAP5b = Tartrate-resistant Acid Phosphatase isoform 5b; P1NP = Pro-collagen Type I N-propeptide; IGF-1 = Insulin-like Growth Factor-1; ACTH = Adrenocorticotropic Hormone; DKK1 = Dickkopf-related protein 1; OPG = Osteoprotegerin; OCN = Osteocalcin; OPN = Osteopontin; SOST = Sclerostin; PTH = Parathyroid Hormone; FGF-23 = Fibroblast Growth Factor-23; BMD = bone mineral density; TGF-β1 = Transforming Growth Factor β1; RCT = randomised controlled trial.

**Table 4 nutrients-17-03845-t004:** Summary of studies assessing the effect of different diets on vitamin D and calcium status.

Study (First Author, Year)	Design/Population	Dietary Intervention/Comparator	Outcome	Mean (Intervention)	Mean (Control)	Key Findings
Jennings et al., 2018 [28]	RCT, adults	Mediterranean diet vs. control	25(OH)D (ng/mL)	5.2 (1.7–8.8)	3.8 (0.7–6.9)	Slightly higher 25(OH)D in intervention group.
Byberg et al., 2016 [49]	Prospective cohort, adults	Mediterranean diet adherence vs. control	Vitamin D intake (mg/day) Calcium intake (mg/day)	5.56 1298	5.58 1254	No difference in dietary vitamin D intake. Marginally higher calcium intake with Mediterranean diet.
Hinton et al., 2012 [38]	RCT, adults	Calorie restriction vs. control	Vitamin D (ng/mL) Calcium intake (mg/day)	6 ± 7.5 20 ± 210	5.5 ± 6.8 15 ± 200	Slightly higher vitamin D in CR group. Slightly higher calcium intake in CR group.
Villareal et al., 2015 [46]	RCT, older adults	Calorie restriction (CR) vs. ad libitum (AL)	25(OH)D (ng/mL)	+1.9 (0.7); 29.6 (0.6)	−0.3 (0.9); 29.3 (0.9)	Increased 25(OH)D from baseline in CR group.
Pop et al., 2015 [42]	Observational, adults	Dietary exposure not specified	25(OH)D (nmol/L)	68.0 ± 24.2	65.9 ± 17.8	No difference between groups.

**Abbreviations:** 25(OH)D = 25-hydroxyvitamin D; CR = calorie restriction; AL = ad libitum; RCT = randomised controlled trial.

## Data Availability

Not applicable.

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
