# Peer review of "Impact of Dietary Patterns on Skeletal Health: A Systematic Review and Meta-Analysis of Bone Mineral Density, Fracture, Bone Turnover Markers, and Nutritional Status"

_nutrients, 2025, doi:10.3390/nu17243845_

Round 1

Reviewer 1 Report

Comments and Suggestions for Authors

This is a unique comparison of popular diet categories on bone outcomes.  Authors were rigorous in their approach using appropriate tools including PRISMA reporting guidelines, 3 independent reviewers, random effects models, funnel plots, etc.  The number of studies seemed sufficiently large though within some diet categories, the numbers became small.  The manuscript is well written and easy to follow.

Author Response

Reviewer 1

Comment: This is a unique comparison of popular diet categories on bone outcomes.  Authors were rigorous in their approach using appropriate tools including PRISMA reporting guidelines, 3 independent reviewers, random effects models, funnel plots, etc.  The number of studies seemed sufficiently large though within some diet categories, the numbers became small.  The manuscript is well written and easy to follow.

Response: We thank the reviewer for these very positive comments. Indeed, there is a small number of studies for some of the diets included in this systematic review and this is emphasised in the limitations as well as in the suggestions for future studies that are needed in the field.

Reviewer 2 Report

Comments and Suggestions for Authors

Dear Authors,

Commonly, an interesting review, but I have some minor remarks:

1) Introduction. Please, ad the end the Novelty of this review!

2) Conclusions. Please, remove the last sentence from this part and move it to the end of Discussion.

Author Response

Reviewer 2

Dear Authors,

Commonly, an interesting review, but I have some minor remarks

Comment 1: Introduction. Please, ad the end the Novelty of this review!

Response 1: We thank the reviewer for this suggestion which helps us to highlight the novelty of our work. This is now added in Lines 116-118 of the revised version.

Comment 2: Conclusions. Please, remove the last sentence from this part and move it to the end of Discussion.

Response 2: Thanks for this useful point. The last sentence in the Conclusions has now been placed as the last sentence of the Discussion, as suggested (Lines 635-641).

Reviewer 3 Report

Comments and Suggestions for Authors

 This systematic review and metaanalysis aimed to evaluate the impact of Mediterranean, calorie restriction, high-protein,  low-carbohydrate and ketogenic diets on skeletal outcomes in adults.

The introduction is adequate for presenting the study hypothesis. It might be interesting to include some reference to diets in relation to weight and the consequences for bone health.

The methodology is described in sufficient detail for the study to be replicated by another research group.

The results are clearly described and easy to understand.

The discussion is appropriate to the results obtained. The limitations highlighted include the heterogeneity of the studies and the need for prospective randomised studies with specific objectives.

Author Response

Reviewer 3

Comment 1: This systematic review and meta-analysis aimed to evaluate the impact of Mediterranean, calorie restriction, high-protein, low-carbohydrate and ketogenic diets on skeletal outcomes in adults.

The introduction is adequate for presenting the study hypothesis. It might be interesting to include some reference to diets in relation to weight and the consequences for bone health.

The methodology is described in sufficient detail for the study to be replicated by another research group.

The results are clearly described and easy to understand.

The discussion is appropriate to the results obtained. The limitations highlighted include the heterogeneity of the studies and the need for prospective randomised studies with specific objectives.

Response: We cordially thank the reviewer for these encouraging comments. With regards to reviewer’s point for the weight, we had already made some comments in the Discussion of the original manuscript. For example, we had described the effects of calorie restricted diet (Lines 536-543), and ketogenic diet (Lines 574-577) on weight loss and their association with bone turnover. We believe that these references address this important aspect which the reviewer kindly highlighted.

Reviewer 4 Report

Comments and Suggestions for Authors

The manuscript “Impact of Dietary Patterns on Skeletal Health: A Systematic Review and Meta-Analysis of Bone Mineral Density, Fracture, Bone Turnover Markers, and Nutritional Status” present an interesting well-documented review.

The aim of the review is the impact of nutrition on skeletal health no data about musculoskeletal and osteoporosis are presented. Consequently, the two first paragraph are not adapted and should be suppressed from the introduction Relation between diet and musculoskeletal diseases and osteoporosis should be presented in the discussion.

As bone, mineral density is usually stable from the end of the growth (18 years) to old age, menopauses for the women and about 60 years for the men. It is not surprising, that in absence of pathologies if the population analyzed is adult over 18 years, the correlation between diet and bone health were not easy to establish.  However, the discussion is very interesting and well present why it is difficult to established correlation between diet and bone health. The authors also well put in light what can be done to improve the understanding of the relationship between nutrition and bone health. A better definition of parameters such as selection of only older adults and improvement of the methodology to evaluate bone quality are needed. Moreover, when we evaluate the impact of a diet, the protein intake and the quality of the protein ingested (level of indispensable amino acids) are important parameters that should be considered.

Author Response

Reviewer 4

Comment 1: The manuscript “Impact of Dietary Patterns on Skeletal Health: A Systematic Review and Meta-Analysis of Bone Mineral Density, Fracture, Bone Turnover Markers, and Nutritional Status” present an interesting well-documented review.

Response 1: We sincerely thank the reviewer for this comment.

Comment 2: The aim of the review is the impact of nutrition on skeletal health no data about musculoskeletal and osteoporosis are presented. Consequently, the two first paragraph are not adapted and should be suppressed from the introduction Relation between diet and musculoskeletal diseases and osteoporosis should be presented in the discussion.

Response 2: We are grateful for this comment. We agree that this systematic review does not analyse any data from studies including osteoporotic patients. However, as the majority of the studies included participants with an age >50, where the fracture risk dramatically increases in both men and women (see PMID: 11095169, PMID: 9844110, PMID: 26968752), we feel that the reference to the most prevalent MSK disease, i.e. osteoporosis, supports the context and rationale of our study as well as directs the integration of our subsequent meta-analysis. Therefore, this part of the Introduction helps the reader to understand the association between nutrition and skeletal homeostasis and how specific diets can have an impact on bone health (e.g. Mediterranean diet on fracture risk) in the general population. We believe that the first two paragraphs are inextricably linked to the conceptualisation of the study and their location in the Introduction contributes to the logical flow of the text.

Comment 3: As bone, mineral density is usually stable from the end of the growth (18 years) to old age, menopauses for the women and about 60 years for the men. It is not surprising, that in absence of pathologies if the population analyzed is adult over 18 years, the correlation between diet and bone health were not easy to establish.  However, the discussion is very interesting and well present why it is difficult to established correlation between diet and bone health. The authors also well put in light what can be done to improve the understanding of the relationship between nutrition and bone health. A better definition of parameters such as selection of only older adults and improvement of the methodology to evaluate bone quality are needed. Moreover, when we evaluate the impact of a diet, the protein intake and the quality of the protein ingested (level of indispensable amino acids) are important parameters that should be considered.

Response 3: We appreciate this reviewer’s comment. We agree with the reviewer that between 20-50, bone mass has plateaued, however the deterioration of bone mass starts (with a slow rate) as early as 45 years of age (please see PMID: 35662432, PMID: 40972625) and, therefore, a nutritional intervention at this point could contribute to the prevention of future fractures. To highlight the older age of the population used in our work, we added this point in Line 218 of the revised manuscript. We had already addressed the reviewer’s comment for the methodology to evaluate bone quality (Lines 621-624 in the revised version), and we hope that this satisfies the reviewer. Finally, we have added Lines 560-563 in the Discussion to address the reviewer’s comment for the protein intake, as kindly suggested.